# Time-resolved cryo-EM visualizes ribosomal translocation with EF-G and GTP

Christine E. Carbone[1], Anna B. Loveland[1], Howard B. Gamper Jr [2], Ya-Ming Hou [2], Gabriel Demo [1,3✉] &
Andrei A. Korostelev [1✉]

During translation, a conserved GTPase elongation factor—EF-G in bacteria or eEF2 in eukaryotes—translocates tRNA and mRNA through the ribosome. EF-G has been proposed to act as a flexible motor that propels tRNA and mRNA movement, as a rigid pawl that biases unidirectional translocation resulting from ribosome rearrangements, or by various combinations of motor- and pawl-like mechanisms. Using time-resolved cryo-EM, we visualized GTP-catalyzed translocation without inhibitors, capturing elusive structures of ribosome•EF-G intermediates at near-atomic resolution. Prior to translocation, EF-G binds near peptidyl-tRNA, while the rotated 30S subunit stabilizes the EF-G GTPase center. Reverse 30S rotation releases Pi and translocates peptidyl-tRNA and EF-G by ~20 Å. An additional 4-Å translocation initiates EF-G dissociation from a transient ribosome state with highly swiveled 30S head. The structures visualize how nearly rigid EF-G rectifies inherent and spontaneous ribosomal dynamics into tRNA-mRNA translocation, whereas GTP hydrolysis and Pi release drive EF-G dissociation.

[1] RNA Therapeutics Institute, UMass Chan Medical School, Worcester, MA, USA. [2] Department of Biochemistry and Molecular Biology, Thomas Jefferson University, Philadelphia, PA, USA. [3] Central European Institute of Technology, Masaryk University, Kamenice 5, Brno 625 00, Czech Republic. ✉email: gabriel.demo@ceitec.muni.cz; Andrei.Korostelev@umassmed.edu

Continuous protein synthesis depends on the synchronous translocation of mRNA and tRNAs through the ribosome (reviewed in refs. [1–3]). After peptide bond formation, the pre-translocation ribosome contains peptidyl-tRNA in the A (aminoacyl-tRNA) site and deacyl-tRNA in the P (peptidyl-tRNA) site, which must be translocated with their mRNA codons to the P and E (exit) sites, respectively (Fig. 1a). The pre-translocation ribosome samples two globally different conformations, which interconvert spontaneously. These are the non-rotated and rotated conformation, in which the small subunit is rotated by up to 10°[4,5]. In the rotated ribosome, the tRNA anticodon stem loops (ASLs) remain bound to the mRNA codons in the A and P sites on the small subunit, while the acceptor arms of tRNAs are shifted into the P and E sites of the large subunit[6–9], thus adopting hybrid states denoted as A/P peptidyl-tRNA and P/E deacyl-tRNA[10]. In the next translocation step, the ASLs and mRNA shift along the small subunit, forming a post-translocation ribosome—with P-site peptidyl-tRNA and E-site deacyl-tRNA—prepared to accept the next aminoacyl-tRNA and continue the elongation cycle[11].

Translocation of the ASLs and mRNA along the small ribosomal subunit is catalyzed by a conserved GTPase, elongation factor G (EF-G) in bacteria or EF-2 in archaea and eukaryotes (Fig. 1a). The structural mechanism of translocation has not been visualized because the rapid GTP hydrolysis step has prevented the capture of authentic EF-G-bound structural intermediates. Prior studies relied on stalling EF-G on the ribosome by antibiotics[12–15], EF-G mutations[16,17], or non-hydrolyzable GTP analogs[18,19], which might capture off-pathway states[20]. Structural studies captured ribosome•EF-G conformations ranging from rotated pre-translocation-like[12] through mid-rotated[13,14,19] to non-rotated post-translocation-like[15] or non-rotated pre-translocation-like states[17]. The structural relationship between GTP hydrolysis, EF-G rearrangements, and translocation, however, remains uncharacterized, as some stalled structures may be inconsistent with the biochemical progression of translocation. For example, a crystallographic pre-translocation-like ribosome structure captured mutant EF-G with GDP[17], whereas in solution, pre-translocation ribosomes bind EF-G•GTP[21,22]. Furthermore, post-translocation ribosomes were reported with GTP-bound-like conformations of mutant EF-G or of EF-G with GTP analogs[16,18,19], whereas authentic post-translocation states must feature post-GTP-hydrolysis states of EF-G.

Two groups of mechanistic models, as well as their combinations, have been suggested to explain EF-G•GTP-catalyzed translocation. In the first group of mechanisms, the energy of GTP hydrolysis is proposed to directly contribute to translocation[3,23] by causing a large-scale conformational change of EF-G[17,24] to exert force[25,26] and/or by inducing ribosome rearrangements that propel tRNA movement[20,27,28]. A ribosome crystal structure with a compact EF-G mutant fused with L9 suggested a nearly 100-Å inter-domain movement[17] toward an extended EF-G conformation captured in most structural studies, in keeping with EF-G acting as a flexible motor. The second group of mechanistic models argues that EF-G acts as a steric hindrance, or pawl, that rectifies the inherent thermal motions of the ribosome, including spontaneous interconversion between non-rotated and rotated conformations, into tRNA translocation[29]. These models are

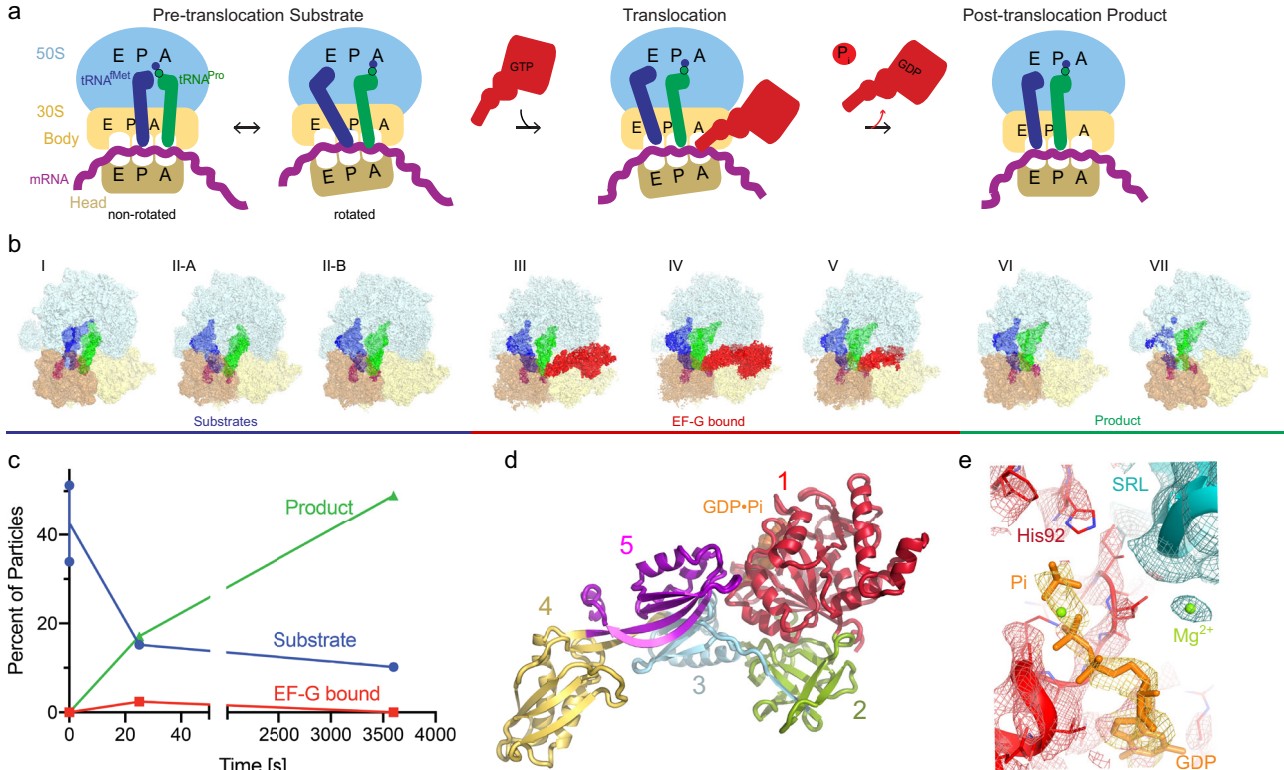

**Fig. 1 Time-resolved cryo-EM of translocation with EF-G and GTP. a** Scheme of the translocation reaction of the 70S•mRNA•fMet-tRNA^fMet•Pro-tRNA^Pro complex with EF-G•GTP. **b** Segmented cryo-EM maps of 8 states of the translocation reaction, and their assignment as substrates, EF-G-bound intermediates, or products of the reaction. The maps are colored to show the 50S ribosomal subunit (light blue), 30S ribosomal subunit body (yellow) and head (tan), tRNA^fMet (dark blue), tRNA^Pro (green), mRNA (magenta) and EF-G (red). **c** Relative abundance of substrates (blue), EF-G intermediates (red), and translocation products (green) over time, obtained from particle distributions in cryo-EM datasets. **d** Domain organization of EF-G; Arabic numerals denote the five conserved domains of the elongation factor. **e** Cryo-EM density of the EF-G GTPase center in the transient pre-translocation and pre-Pi-release state (III). For additional density views, see Fig. 4 and Supplementary Figs. 3, 4, and 5.

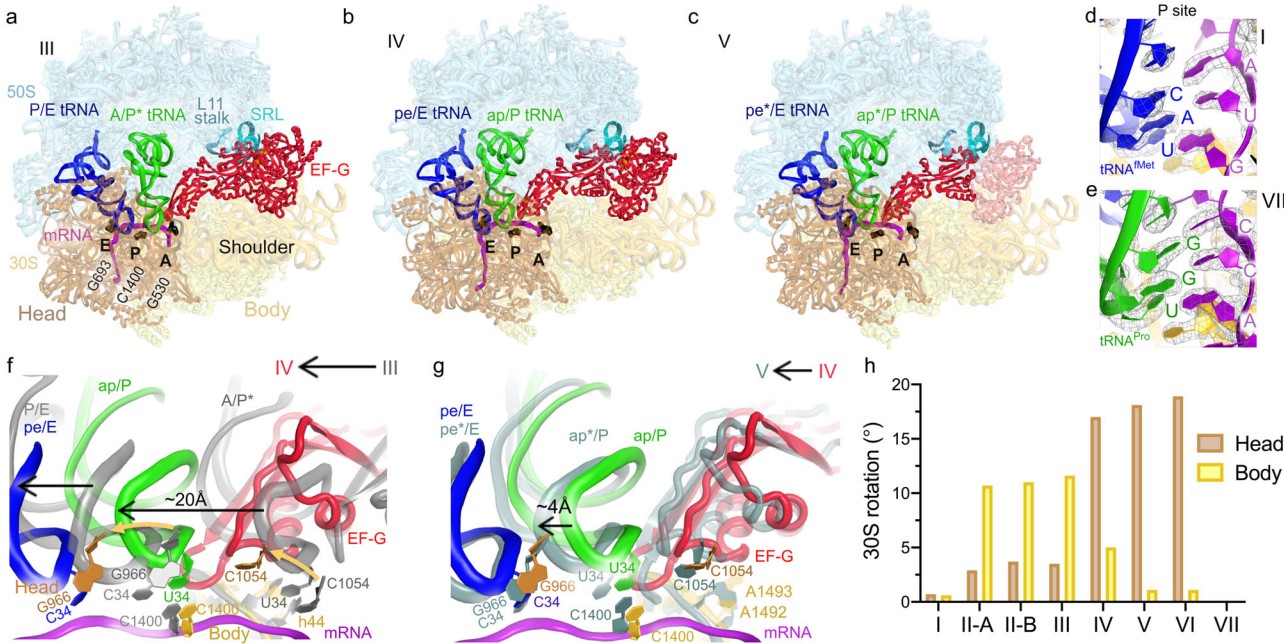

**Fig. 2 Structures of translocation intermediates with EF-G. a–c** Structures III, IV and V with EF-G. 16S nucleotides at the A, P, and E sites (G530, C1400 and G693, respectively) are shown as black surfaces for reference. Unresolved part of EF-G in Structure V is shown in transparent red in panel c. **d–e** high-resolution density identifying pre-translocation (**d**) and post-translocation (**e**) tRNA anticodon and mRNA codon in the P sites of Structures I and VII. **f–g** Transitions of tRNA and EF-G between Structures III and IV (**f**) and Structures IV and V (**g**). **h** Degrees of 30S head swivel and body rotation in Structures I through VII.

consistent with the ribosome's ability to slowly translocate tRNA in the absence of EF-G[30–33], indicating that translocation is an inherent property of the ribosome. Because non-hydrolyzable GTP analogs efficiently catalyze translocation, accelerating it by more than 10³-fold[11,23,27,34], the translocation stage was proposed to be independent of GTP hydrolysis and large-scale interdomain rearrangements of EF-G[12]. Nevertheless, the rate of translocation in the presence of EF-G and GTP is up to 50-fold higher than with GTP analogs or catalytically inactive EF-G[11,16,23,27,35]. Thus, neither group of models fully explains the structural roles of EF-G and GTP hydrolysis[1,3].

To understand how EF-G and GTP catalyze translocation, we performed time-resolved cryogenic electron microscopy to visualize authentic translocation intermediates without inhibitors (Fig. 1 and Supplementary Figs. 1–2). We report three ribosome intermediates with EF-G (Fig. 1a–b), resolving EF-G's GTPase center at ~3.5 Å local resolution (Fig. 1h and Supplementary Fig. 3). Together with the pre-translocation and post-translocation states observed without EF-G, our data allow reconstruction of the structural pathway of translocation, elucidating the structural roles of EF-G (Figs. 2 and 3) and GTP hydrolysis (Fig. 4). In addition to inhibitor-free complexes, we report a 3.2-Å structure of a pre-translocation complex formed with EF-G•GTP and stalled with viomycin, which supports our findings (Supplementary Figs. 2 and 3). Comparison with previous structural studies suggests that some conformations of EF-G or ribosomes stalled by GTP analogs, antibiotics, and mutations, may represent the states that are lowly-populated or not populated during GTP-catalyzed translocation.

## Results

**Cryo-EM captures EF-G•GTP translocation intermediates.** To visualize EF-G-catalyzed translocation using time-resolved cryo-EM, we added *E. coli* EF-G•GTP to pre-translocation 70S ribosomes with tRNA^fMet in the P site and with dipeptidyl-tRNA^Pro in the A site programmed with the cognate CCA codon (Fig. 1a;

Methods). The reaction was performed on ice to slow translocation[32] and enable capturing translocation intermediates. We plunged EM grids into a cryogen to stop the reaction at 0 (prior to adding EF-G•GTP), 25, and 3600 s, and collected cryo-EM data for each time point. Maximum-likelihood classification of ribosome particles in these datasets identified eight structures, comprising three major functional states of the ribosome: pre-translocation substrates, EF-G-bound intermediates, and post-translocation products (Fig. 1b and Supplementary Fig. 1). As the population of pre-translocation ribosomes decreases over time, the population of post-translocation ribosomes increases, as expected. EF-G-bound intermediates were only observed in data from the intermediate time point (Fig. 1c).

The pre-translocation substrates, obtained without EF-G, feature three conformations of the ribosome and tRNAs: non-rotated Structure I and rotated Structures II-A and II-B (Fig. 1b and Supplementary Fig. 4). Structure I contains tRNAs in the classical A (A/A) and P (P/P) states, and a low-occupancy E-site (Supplementary Fig. 4a), similar to those in previous studies of elongation (e.g. Structure V-B in ref. [36]). Structure II-A features a rotated ribosome with an A/A-like tRNA^Pro and P/E tRNA^fMet (Supplementary Figs. 4b–c and 5a–b). This state reveals that upon intersubunit rotation, dipeptidyl-tRNA can remain in the 50S A site while the acceptor arm of the deacyl-tRNA shifts to the E site, as suggested by studies using Fluorescence/Förster Resonance Energy Transfer (FRET)[37] and mutant bacterial ribosomes[38]. Structure II-B features a rotated ribosome with an A/P* and P/E hybrid-state tRNAs, whose acceptor arms are shifted to the P and E sites on the large subunit, respectively (Supplementary Figs. 4d and 5a–b)[6,9]. The elbow of A/P* tRNA is shifted by ~25 Å toward the P site relative to the canonical A/P tRNA observed in most studies with different tRNAs. Our data also contain ribosomes with weak density for A/P tRNA, although we could not unambiguously separate them into a high-resolution class with A/P tRNA. Overall, conformations of tRNA^Pro are similar to those sampled by other tRNA species (e.g. tRNA^Phe) on

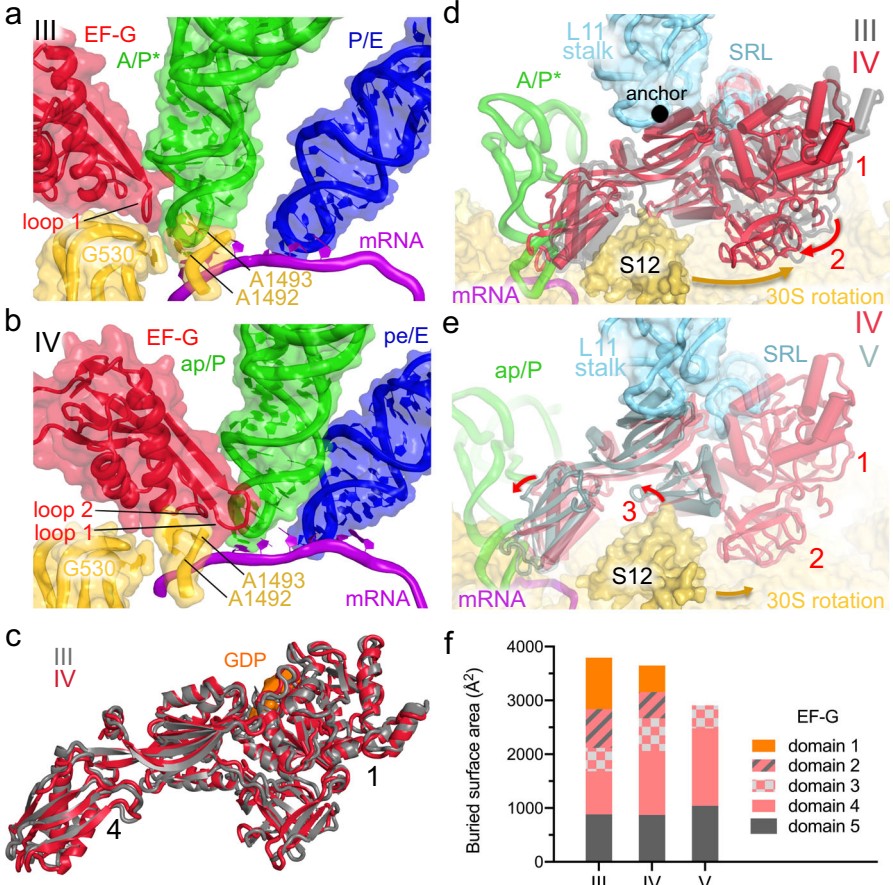

**Fig. 3 Positions and interactions of EF-G in translocation intermediates. a–b** Positions of EF-G and tRNAs relative to the decoding center (yellow) in Structures III and IV. **c** Superposition of EF-G in Structures III and IV demonstrates an overall similar extended conformation. **d** Movement of EF-G relative to the 30S subunit from Structure III (gray) to IV (colored). Structures are aligned on 16S rRNA and colored ribosomal elements are from Structure III. **e** Movement of EF-G relative to the 30S subunit from Structure IV (colored) to V (blue-gray). Structures are aligned on 16S rRNA and colored ribosomal elements are from Structure IV. **f** Buried surface area (contact area) showing the extent of interactions of EF-G domains with the ribosome, mRNA, and/or tRNA in Structures III to V.

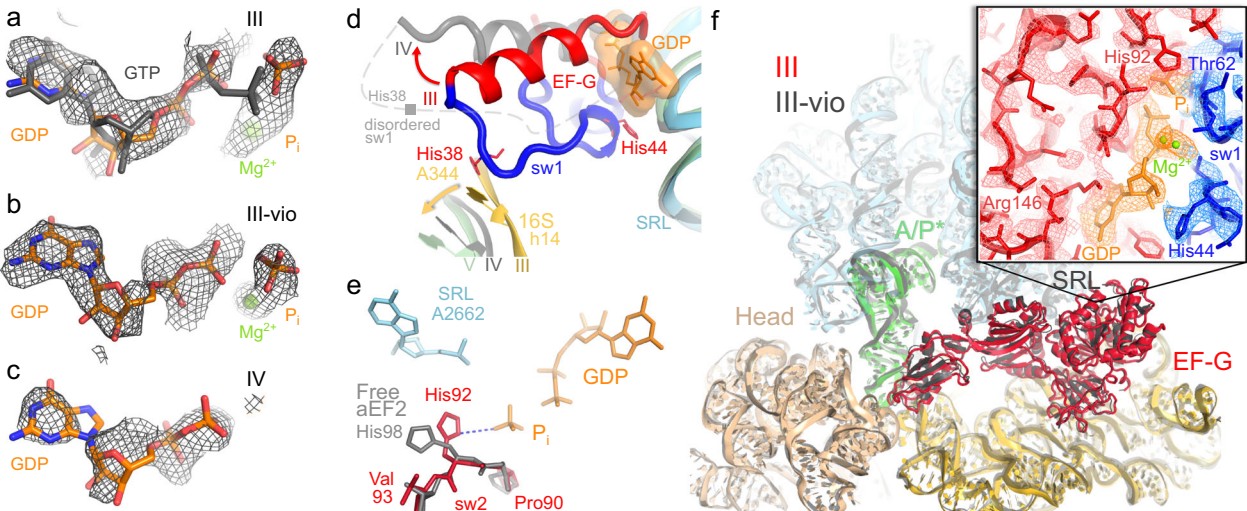

**Fig. 4 The GTPase center of EF-G in translocation Structures III, III-vio and IV. a–c** Cryo-EM densities are consistent with GDP•Pi in Structures III and III-vio, and with GDP in Structure IV. Grey model shows GTP for reference (1WDT). **d** Positions of sw-I in Structure III (ordered) and IV (gray, disordered) between the SRL and h14 of the 30S subunit. **e** The catalytic conformation of His92 in Structure III (colored) differs from position of this side chain in the off-ribosome EF-G homolog (archaeal EF-2, gray) bound with GTP analog[57]. **f** Pretranslocation 70S•EF-G•GDP•Pi structure captured with viomycin (III-vio; gray) is nearly identical to Structure III. Box shows density for EF-G GTPase center in Structure III-vio (His92 density is shown in Supplementary Fig. 2c).

pre-translocation ribosomes from bacteria[36] and eukaryotes[39,40], in keeping with the conservation of ribosomal and tRNA rearrangements during the elongation cycle. Our Structures I through II-B confirm that prior to translocation, the ASL of peptidyl-tRNA remains in the 30S A site, whereas the acceptor arm and elbow spontaneously sample different conformations, including the A/P* state, which is most advanced toward the P site.

Three structures of transient translocation intermediates (III, IV, and V) include long-sought EF-G-bound states resolved at ~3.5 Å average resolution. These structures represent distinct stages of tRNA advancement along the translocation trajectory (Fig. 2). Structure III features a pre-translocation 70S•EF-G complex with A/P* and P/E tRNAs (Fig. 2a). In Structure IV, EF-G is shifted ~20 Å along the 30S subunit, with partially translocated "chimeric" ap/P and pe/E tRNAs (Fig. 2b; tRNA nomenclature as in[14]). In Structure V, tRNA ASLs are further along the translocation pathway, reaching the P and E sites of the 30S body domain (we term the tRNAs ap*/P and pe*/E; Fig. 2c). EF-G translocase domain shifts with the tRNAs by another 4 Å, whereas the N-terminal domains 1 and 2 are characterized by scattered low-resolution density (Fig. 2c; Supplementary Figs. 3h and 4g), indicating that they are dynamic in the intersubunit space.

Two post-translocation products (VI and VII) lack EF-G. The ribosome with a highly swiveled 30S head and ap*/P and pe*/E tRNAs in Structure VI is nearly identical to the EF-G-bound Structure V. Structure VI is found exclusively in the 25-s data set, indicating that it is a transient state formed after EF-G dissociation. By contrast, the terminal post-translocation Structure VII with fully translocated dipeptidyl-tRNA^Pro in the P site is present in both the 25- and 3600-s data sets. Here, the non-rotated ribosome with a non-swiveled head is nearly identical to the ribosome in Structure I (Fig. 2h) and previous structures of post-translocation complexes, and the ribosome is ready for the next round of elongation.

**Extended EF-G binds rotated pre-translocation ribosomes with A/P* tRNA.** During translocation, the ASL of peptidyl-tRNA with its mRNA codon must traverse a ~25 Å distance from the A to P site (measured between anticodon nucleotide-34 phosphates of A- and P-site tRNAs). This movement is accompanied by the reverse ~10° rotation of the 30S subunit body[11]. Moreover, ribosome structures with inhibitors[14,41–43], and solution FRET studies[7,44], reported that EF-G or eEF-2 binding is associated with a large, up to ~20°, rotation ("swivel") of the head domain of the small subunit. However, authentic EF-G-bound translocation intermediates with rotated body or swiveled head have eluded structural characterization.

Structure III (Fig. 2a) reveals EF-G bound to a fully rotated ribosome with A/P* tRNA, similar to pre-translocation Structure II-B captured prior to EF-G binding (Fig. 2h and Supplementary Table 2). Superposition with other rotated pre-translocation states reveals that EF-G is less compatible with tRNAs in the A/A (Structure II-A) and A/P conformations (PDB 6WDF; Supplementary Fig. 5e–f), as they would clash with the EF-G translocase domain 4 (aa 490–610; E.coli numbering; Arabic numerals are used for EF-G domain designation as in ref. [45]). These observations indicate that the spontaneously sampled ribosome with A/P* tRNA is a likely substrate for EF-G binding.

In Structure III, EF-G adopts an extended conformation in the intersubunit space, spanning 100 Å from the GTPase domain to the tip of domain 4 (Figs. 1d, 2a). The GTPase domain (domain 1, also termed the G-domain; aa 1–290; Fig. 1d) binds at the universally conserved sarcin-ricin loop of the large subunit (SRL; nucleotides 2653–2667 of 23S rRNA). In its vicinity, EF-G

domain 5, which forms the translocase superdomain with domain 4, binds to the L11 stalk (23S residues 1050–1105 and protein uL11; Fig. 2a and Supplementary Fig. 6a). Domains 2 and 3 (part of the GTPase superdomain) bind at a peripheral region of the 30S shoulder and body domains (Fig. 2a).

EF-G domain 4 is inserted between dipeptidyl-tRNA^Pro, the 30S shoulder and the 30S head domains (Fig. 2a). Loop 1 at the tip of domain 4 (aa 507–514) is wedged between the tRNA and decoding center, where it reaches toward the 16S nucleotide G530 on the 30S shoulder (Fig. 3a and Supplementary Fig. 6b–c), one of the three universally conserved nucleotides critical for mRNA decoding and A-site-tRNA stabilization or locking[46–48]. The neighboring loop 2 (aa 582–588) fits into a minor groove of helix 34 of 16S rRNA (at C1209), binding the 30S head in a pre-swiveled conformation. Thus, domain 4 is positioned to separate, or unlock, the codon-anticodon helix from the decoding center and follow the head during translocation.

**Nearly rigid EF-G enters the A site during reverse 30S body rotation.** In the mid-translocation Structure IV, EF-G retains an extended conformation, similar to that in Structure III, with EF-G domain 5 anchored to the L11 stalk of the 50S subunit (Fig. 2b). Because the 30S body has reversed its rotation to 5.0° (from 11.6° in Structure III, Supplementary Table 2), EF-G domains 2 to 4 have moved along the 30S subunit (Supplementary Fig. 6d–e). Domain 4 is shifted 20-Å (measured at Tyr 515 near loop 1) from its position in Structure III, inserting into the A site (Figs. 2f and 3b). Loop 1 has moved to contact nucleotides of both the CCA codon and the dipeptidyl-tRNA^Pro anticodon (Supplementary Fig. 6f–g). In this position, domain 4 separates the tRNA from the decoding-center nucleotides (Fig. 3b). The ASL of dipeptidyl-tRNA^Pro is translocated 18-Å relative to the 30S body, so that the anticodon nucleotide U34 stacks on the 30S P-site residue C1400 of 16S rRNA (Fig. 2f). However, due to a 17° swivel of the 30S head toward the large subunit (i.e., in the direction of translocation), dipeptidyl-tRNA remains near the A site of the 30S head, and deacyl-tRNA^fMet remains near the P-site of the 30S head, where C34 stacks on G966 of 16S rRNA (Fig. 2f and Supplementary Fig. 6e). These tRNA conformations closely resemble chimeric ap/P and pe/E tRNAs captured in the presence of fusidic acid or neomycin[13,14]. Unlike in the neomycin-bound high-resolution structure, in which the tRNA acceptor arm is between the 50S A and P sites[13], Structure IV features the acceptor arm of the ap/P tRNA bound to the P loop of the 50S P site (Supplementary Fig. 5c–d).

To accommodate into the A site, EF-G undergoes small-scale rearrangements (Fig. 3c), as domain 4 shifts relative to the GTPase domain by ~7 Å (RMSD, root-mean-square distance between superimposed EF-G from Structures III and IV). The range of interdomain rearrangements is similar to or less than interdomain fluctuations of free EF-G in solution[49] and in crystal structures of free EF-G homologs[45,50,51] (up to ~20 Å; Supplementary Fig. 7), suggesting that EF-G undergoes local stochastic rearrangements to accommodate into the A site during translocation.

Structure V represents a heretofore unseen EF-G-bound ribosome state with a highly swiveled head and further translocated tRNAs (Fig. 2c, g). 30S body is less rotated (1.1°) than that in Structure IV, whereas head swivel (18.1°) is slightly increased (Fig. 2h). EF-G domain 4 and tRNAs have advanced 3–5 Å along the 30S subunit. The ASL of dipeptidyl-tRNA^Pro is placed deeper into the P site of the 30S body, forming a late translocation state ap*/P (Fig. 2g and Supplementary Fig. 6h). Strong density shows EF-G domain 4 occupying the ribosomal A site and domain 5 attached to the L11 stalk. Density for domain

3 is weaker, and densities for domains 1 and 2 are non-continuous and low-resolution (Fig. 3e and Supplementary Figs. 3d, h and 4g). Thus, Structure V is consistent with a late translocation intermediate, in which EF-G releases its hold on the ribosome, as the GTPase domain leaves the SRL and domain 2 leaves the 30S subunit. Dissociation of EF-G domains 1 and 2 correlates with steric hindrance presented by ribosomal protein uS12 (bacterial S12; Fig. 3e) and loss of interactions between domain 1 and the back-rotating 30S subunit, as discussed below. Step-wise dissociation of EF-G resembles that of EF-Tu, whose GTPase domain is released from the ribosome before other domains during tRNA decoding[36].

Structural analyses of EF-G-bound intermediates highlight that progression from Structure III to V is correlated with a stepwise loss of EF-G contact with the ribosome (Fig. 3f). Extensive interactions of the GTPase with the SRL and the small subunit in Structure III (GTPase-domain buried surface area of ~960 Å$^2$) are halved in Structure IV (493 Å$^2$) on the path to dissociation of the GTPase domain in Structure V (~0 Å$^2$). By contrast, the translocase superdomain expands its interactions with the ribosome. The invariant interaction of EF-G domain 5 with the L11 stalk of the 50S subunit in all three structures (~900 Å$^2$) holds EF-G in place to allow the entry into the A site during reverse 30S rotation. Interactions of domain 4 with the 30S subunit gradually expand from 800 Å$^2$ in Structure III through 1190 Å$^2$ in Structure IV to 1440 Å$^2$ in Structure V (Fig. 3f). Nevertheless, the overall contact area of EF-G during transloca-tion reduces from 3730 Å$^2$ (whole EF-G buried surface area in Structure III) to 3530 Å$^2$ (IV; 95% from that in Structure III) to 2902 Å$^2$ (V; 77% from that in Structure III). Because the buried surface area positively correlates with the binding affinity of macromolecules[52], these measurements suggest that gradual dissociation of EF-G is driven by different affinities of EF-G to the ribosome in different 30S rotation/swivel states.

Structure VI lacks EF-G, but the tRNA positions and 30S conformation only marginally differ from those in Structure V (Fig. 1b and Supplementary Fig. 4h). With a slightly more swiveled head (18.9°), Structure VI represents a transient translocation intermediate following EF-G dissociation.

The completion of tRNA and mRNA translocation along the head requires an ~20° reversal of head swivel, to the post-translocation state captured in Structure VII (Fig. 1b and Supplementary Fig. 4i). The non-rotated/non-swiveled Structure VII features an empty A site and tRNA$^{Pro}$ with the associated proline codon clearly resolved in the P site (Fig. 2d–e). Very low density suggests that the bulk of deacyl-tRNA$^{fMet}$ has dissociated from the E site (Fig. 1b and Supplementary Fig. 4i). Extensive classification of cryo-EM data did not detect EF-G on non-rotated, post-translocation ribosomes that would resemble EF-G-bound structures stalled by fusidic acid[15], GTPase-defective EF-G mutant[16] or non-hydrolyzable GTP analog[18,19]. Our structures therefore suggest that in the absence of inhibitors, EF-G dissociates before or during reversal of head swivel.

**Pi release during tRNA translocation.** Structures III and IV reveal two functional EF-G GTPase states distinguished by dif-fering interactions with the 30S subunit and the SRL, which is essential for the hydrolysis of GTP[15,53,54]. Structure III features an activated GTPase domain clamped between the 30S and 50S subunits. The EF-G GTPase center is resolved to a local resolu-tion of ~3.5 Å, allowing detailed interpretation of its conforma-tion (Figs. 1e, 4a and Supplementary Figs. 3b, f, j). Two switch loops (sw-I and sw-II) outline the GTP-binding pocket (Supple-mentary Figs. 3 and 6i[45,55]). The longer sw-I (aa 35–65) bridges the SRL with the small subunit, as described below. His44 of sw-I

docks at G2655 of the SRL and stabilizes the ribose of GDP (Fig. 4d). On the opposite wall of the GTP-binding pocket, cat-alytic His92 of sw-II (aa 80–95) docks near the SRL at the phosphate of A2662, with its side chain oriented toward the γ-phosphate (inorganic phosphate, Pi, Fig. 4e) and Ile61 of sw-I, consistent with a catalytically activated GTPase[56]. This con-formation contrasts with pre-GTP-hydrolysis states of free EF-G homologs[51,57], wherein His92 points away from γ-phosphate (Fig. 4e). Nevertheless, the overall fold of the GTPase center is remarkably similar to those in unbound GTPases (Supplementary Fig. 6i), indicating that ribosome binding induces only a local rearrangement of His92 to activate GTP hydrolysis. Strong den-sity suggests that the γ-phosphate is separated from GDP (Fig. 4a) and stabilized by hydrogen-bonding donors and positively charged side chains of sw-I and sw-II (Supplementary Fig. 6j). The relative positions of Pi and GDP are nearly identical to those in high-resolution crystal structures of Ras GTPase and aIF2 (Supplementary Fig. 6i[58]). Moreover, the density is less compa-tible with GTP (Supplementary Table 3), whose covalently bound γ-phosphate would be closer to its β-phosphate (Fig. 4a[51]). Thus, consistent with the fast chemical reaction[32,59], Structure III represents predominantly a post-GTP-hydrolysis step with GDP and Pi stabilized by the ordered switch loops.

Sw-I is sandwiched between the SRL and the 30S subunit, where His38 of sw-I packs on the bulged A344 residue in helix 14 of 16S rRNA (Fig. 4d). Helix 14 is closer to the SRL in the rotated ribosome (28 Å between the phosphates of A344 and A2662) than in the non-rotated conformation (36 Å), indicating that the rotated 30S conformation helps stabilize the active conformation of the GTPase center, leading to GTP hydrolysis (Fig. 4d). Indeed, the rotated pre-translocation ribosome is the authentic substrate for EF-G•GTP binding and hydrolysis[22,60], whereas an isolated SRL RNA oligonucleotide does not activate GTP hydrolysis on EF-G[53]. Moreover, by stabilizing sw-I, the rotated 30S prevents the release of the Pi, ensuring that EF-G does not dissociate prior to translocation.

Overall, Structure III resembles a 7.4-Å cryo-EM structure of an EF-G-bound ribosome stalled by the addition of the antibiotics fusidic acid and viomycin[12]. Viomycin stabilizes the pre-translocation tRNA in the decoding center without inhibiting GTP hydrolysis[61]. The low resolution prevented a detailed structural analysis of the GTPase center in the antibiotic-bound structure. To further resolve the GTPase center in the pre-translocation EF-G state, we used cryo-EM to visualize a pre-translocation 70S complex assembled with EF-G•GTP and viomycin (Supplementary Fig. 2). Remarkably similar to Structure III, the 3.2-Å resolution structure of the viomycin-stalled ribosome (i.e., III-vio; Supplementary Fig. 2d) features a better resolved GTPase center (Fig. 4b, f and Supplementary Fig. 3i). Structure III-vio supports our finding that the pre-translocation ribosome contains post-hydrolysis Pi and GDP (Fig. 4b; Supplementary Table 3) stabilized by the switch loops and ions, likely magnesium[62,63], which coordinate the phosphate groups (Fig. 4f). These findings are similar to those in two cryo-EM studies of ribosomal EF-G complexes with antibiotics published while our manuscript was under review. GDP and Pi were reported in the cryo-EM structure of pretranslocation ribosome stalled with the antibiotic apramycin, which locks the decoding center similarly to viomycin[64]. EF-G-bound ribosome with the antibiotic spectinomycin is also consistent with post-hydrolysis GDP and Pi, although the structural model was reported as GTP[65] (Supplementary Fig. 6l–n; see Methods and Supplementary Table 3). Thus, our structures III and III-vio demonstrate that (a) GTP is hydrolyzed on pretranslocation ribosome, and (b) after hydrolysis, the switch loops of the GTPase center remain well ordered because they are stabilized by the rotated 30S conformation.

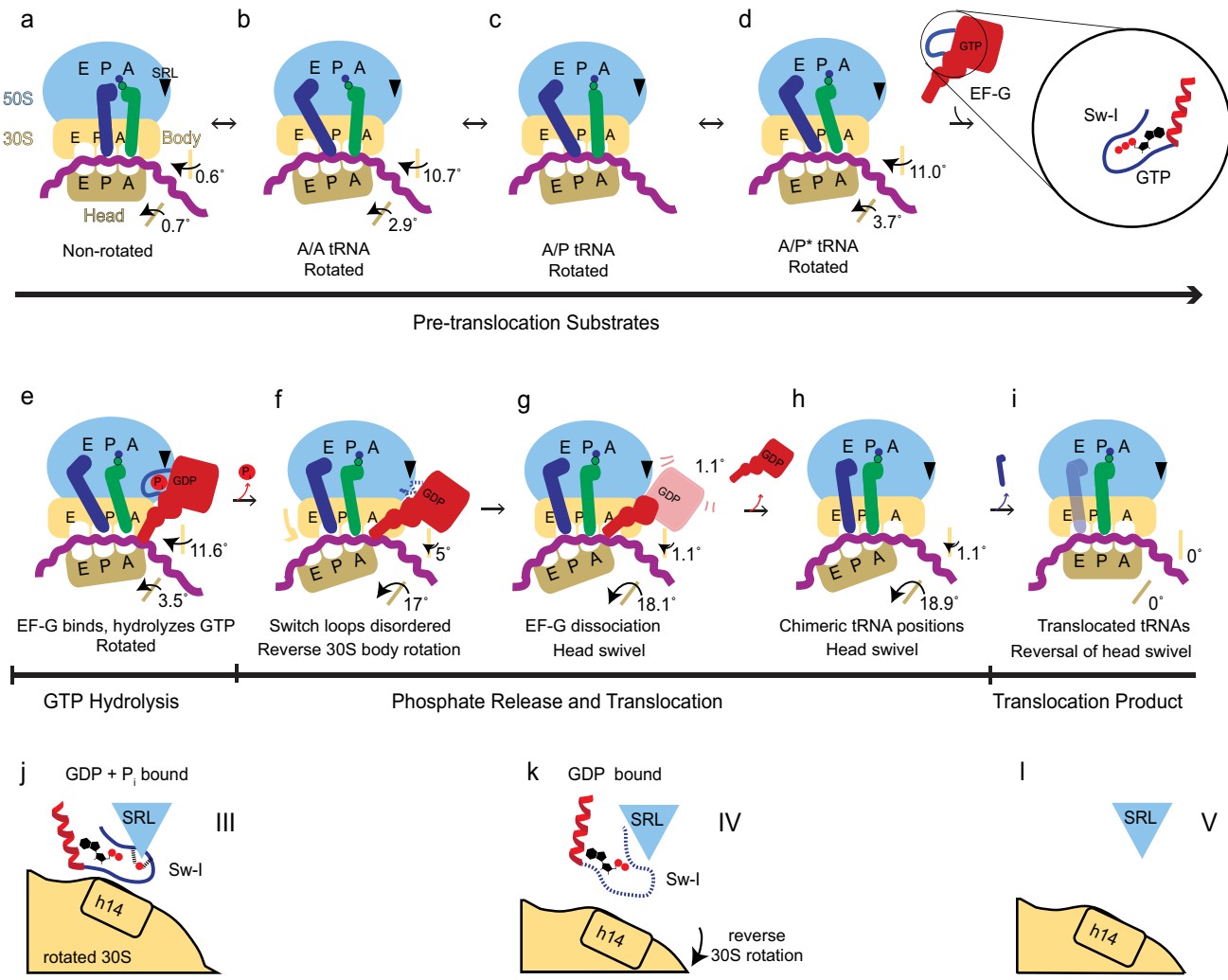

**Fig. 5 Schematic of ribosomal translocation catalyzed by EF-G and GTP. a–i** Progression of translocation and rearrangements of ribosomal subunits, EF-G and tRNAs. **j–l** Rearrangement of EF-G GTPase switch loop I, showing the coupling of Pi release with 30S rotation and translocation.

Structure IV, by contrast, features a post-Pi-release conformation of EF-G. Here, movement of EF-G into the 30S A site coincides with separation of the GTPase domain from the SRL (Fig. 4d). The GTP-binding pocket is ~2 Å further from the catalytic SRL phosphate than in Structure III, consistent with an inactive post-reaction state (Supplementary Fig. 6k). The GTPase movement relative to the SRL is consistent with mutational studies showing that perturbing the conformation of the SRL abolishes translocation even if the GTPase activity is retained[54]. Whereas GDP is clearly resolved, densities for the switch loops and Pi are absent (Fig. 4c–d and Supplementary Fig. 3k), indicating that the switch loops become dynamic and thus release Pi from the GTPase center[59,66]. These rearrangements of EF-G coincide with a >10-Å movement of h14 away from sw-I, as a result of reverse 30S rotation (Fig. 4d). Thus, disruption of the contact between the 30S and GTPase is correlated with Pi release.

## Discussion

**Structural mechanism of EF-G•GTP-catalyzed translocation.** Time-resolved cryo-EM of authentic translocation answers several long-standing questions; rationalizes previous structural, biochemical and biophysical observations; and suggests a parsimonious model for the translocation mechanism (Fig. 5 and Supplementary Movie 1). As Structures I through II-B report, pre-translocation ribosomes spontaneously interconvert between non-rotated and rotated conformations, in which the peptidyl-tRNA samples A/A and A/P* states (Fig. 5a–d). This is consistent with a large body of biochemical and biophysical data reporting fast tRNA fluctuations on the 50S subunit and intersubunit rotation prior to EF-G binding[4,37,67,68]. EF-G•GTP binds to a rotated pre-translocation ribosome[22], where the relative position of the small subunit and the 50S SRL are complementary to the GTP-bound conformation of EF-G's GTPase domain, as in Structure III (Fig. 5e, k). The EF-G translocase domain 4 binds near the ASL of the A/P* tRNA (Fig. 5e). Because EF-G appears sterically incompatible with A/A and A/P tRNA conformations (Supplementary Fig. 5e–f), this binding must shift the conformational equilibrium toward the "elbow-translocated" A/P* conformation. Indeed, in the 25-s dataset with EF-G, no classes of rotated ribosomes with the A/A tRNA (as in Structure II-A) are observed, indicating substantial depletion in comparison with the 0-s dataset (Fig. 1c and Supplementary Fig. 1). The binding of EF-G to the rotated pre-translocation ribosome is consistent with biochemical observations of transient stabilization of the rotated 70S by EF-G with GTP or GTP analogs and increased rates of forward 30S rotation[28,60].

Structures of EF-G-bound intermediates (III through V) report the trajectory of translocation consistent with FRET solution studies showing that translocation on the 30S subunit occurs during the reverse 30S rotation[11] and proceeds in at least two

steps[7,69]. During spontaneous reverse rotation of the 30S subunit, EF-G domain 4 sterically hinders the A/P* tRNA from returning to the canonical A/A state (Supplementary Figs. 5e–f and 6f–h). The tip of domain 4 initially wedges between the tRNA and the shoulder of the 30S subunit (Structure III) and then enters the A site, separating, or unlocking, the tRNA from the decoding center in Structure IV. Loops 1 and 2 of domain 4 interact with the tRNA and with the 30S head (Fig. 3a–b), in keeping with their critical role in tRNA unlocking and 30S head swivel required for translocation[70]. Reverse 30S rotation moves EF-G into the A site first by ~20 Å (from Structure III to IV) and then by ~4 Å (from Structure IV to V), completing translocation relative to the 30S body. The transition of EF-G relative to the A site is consistent with those observed in solution studies[71] and with a recent structure of an EF-G intermediate stalled by a translocation inhibitor spectinomycin[65]. Moreover, EF-G and ribosome rearrangements closely resemble eEF2 and eukaryotic ribosome transitions inferred from an ensemble cryo-EM study of IRES translocation[43]. Thus, the conserved elongation factors induce unlocking and retain contact with peptidyl-tRNA to bias the diffusion of tRNA-mRNA along the mRNA tunnel from the A to P site.

Consistent with FRET studies, the structural intermediates with extensive ~20° head swivel (IV and V), occur on-pathway[7,44] to retain the interactions between the 30S head and translocating tRNAs. Novel states in the 25-s dataset with the most translocated tRNAs and most swiveled 30S head capture a transient equilibrium between the ribosomes with EF-G (Structure V) and without EF-G (Structure VI), suggesting that EF-G•GDP can dissociate from ribosomes prior to the reversal of head swivel (Supplementary Fig. 4g–h). Despite extensive classification, our data have not revealed EF-G bound to non-rotated/non-swiveled ribosomes, suggesting that they are exceedingly rare if they exist during authentic translocation. By contrast, non-rotated/non-swiveled ribosomes with EF-G were reported when EF-G cannot dissociate due to the inability to hydrolyze GTP or due to the presence of an antibiotic[15,16,18]. Our findings therefore illustrate that EF-G•GTP-catalyzed translocation of tRNAs occurs in two major steps: first, relative to the 30S body, coincident with the forward head swivel (with EF-G); second, relative to the head, upon reversal of the head swivel (without EF-G, or coincident with EF-G dissociation).

Our work provides structural insights into the role of GTP hydrolysis in translocation. EF-G accelerates translocation by more than 3 orders of magnitude with either GTP or non-hydrolyzable GTP analogs[27,31,33]. Yet, translocation rates are 2- to 50-fold higher with GTP than with GTP analogs[11,23,27,34] or with inactivating His92 mutations[16,35]. The structural basis for this difference has remained unclear. Our structures demonstrate that rather than being coupled with the chemical step of GTP hydrolysis, tRNA translocation is coupled with switch-loop rearrangements of EF-G and phosphate release (from Structure III to Structure IV). Structures III and III-vio are consistent with biochemical studies, showing that Pi release is slower than hydrolysis and may determine the rate of tRNA translocation[32,59]. In the pre-translocation ribosome (Structure III), sw-I bridges the SRL with the rotated 30S subunit, preserving an ordered GTP-bound-like conformation of the EF-G GTPase center. Reversal of the 30S subunit rearranges sw-I, allowing Pi diffusion from the GTPase center (Structure III to IV). By contrast, artificial prevention of Pi release—e.g., in the presence of non-hydrolyzable GTP analogs or catalytically defective EF-G mutants—stabilizes a GTP-bound-like conformation of EF-G until late translocation states[16,18]. The inability of sw-I to rearrange correlates with the reduced rates of reverse 30S rotation with GTP analogs[11], at least in part explaining the slower

translocation. Moreover, in the presence of non-hydrolyzable GTP analogs, the GTP-like conformation of EF-G prevents the dissociation of the GTPase domain from the SRL at latter stages of translocation[16,18], which coincide with the reversal of head swivel (Structure IV to V to VI). Our structural analyses of EF-G-bound intermediates (Structures III through V) highlight that GTP hydrolysis contributes to the directionality and completion of translocation by enabling a stepwise loss of EF-G contact with the ribosome (Fig. 3f). Indeed, single-molecule FRET and biochemical studies showed that transitions between the late translocation states[7] and dissociation of EF-G[32] may determine the rates of EF-G•GTP-catalyzed translocation.

Due to the strict directionality, large-scale tRNA movements, and fast rates of GTP-catalyzed translocation, some mechanistic models proposed that translocation is driven by large-scale rearrangements of EF-G[17,25,26]. Discussions considered that EF-G could act as a flexible GTPase motor, akin to classic ATP-driven motors[72], such as myosin and kinesin, whose conformational changes are commensurate with their molecular size[73]. The structures captured in this work suggest that EF-G does not act as a highly flexible motor and that the proposed nearly 100-Å rearrangement of domain 4[17] (Supplementary Fig. 7d) is not required for translocation to occur (see Supplementary discussion). EF-G adopts similarly extended conformations in the pre-translocation state before Pi release (Structure III) and in the nearly post-translocated ribosome after Pi release (Structure IV). The ~7 Å displacement of domain 4 from the GTPase domain (Fig. 3c) cannot account for ~25-Å translocation of tRNA and mRNA. Rather, the EF-G interdomain movement is consistent with spontaneous thermal fluctuations of ~10 Å observed in solution studies[49]. Thus, modest interdomain rearrangement of EF-G accounts for accommodation of domain 4 in the 30S A site during reverse 30S rotation. If some large-scale interdomain EF-G rearrangements occur on the ribosome[25], they must take place prior to formation of the pre-translocation Structure III and thus do not drive translocation. By contrast, the 30S body rotation[4,5] and head swivel[41,74–76] are the inherent and spontaneously sampled properties of the ribosome, which have been observed without EF-G. The rates of intersubunit rotation are directly coupled to the rates of translocation[11,77], indicating that ribosomal rearrangements are the driver of translocation. Thus, EF-G accelerates translocation by acting as a nearly-rigid steric block (i.e., a pawl), that rectifies inherent ribosomal rearrangements into tRNA movement on the 30S subunit. The GTPase activity serves as a switch controlling the ability of EF-G to bind and leave the ribosome (Fig. 5j–i).

The translocation intermediates captured in this work also illustrate how the mRNA frame is preserved to prevent frameshifting events that could produce toxic proteins and premature termination. While the pre-translocation ribosome stabilizes the tRNA-mRNA helix in the decoding center and in the P site[46–48], the thermodynamically labile three-base pair codon-anticodon helix may be destabilized during the transition between these two sites, leading to tRNA slippage and frameshifting. Indeed, a recent crystal structure revealed that the tRNA-mRNA base-pairing can be destabilized in the absence of EF-G, if the 30S body and head are rotated similarly to those in our EF-G-bound Structure IV[75] (Supplementary Fig. 8d). We also recently reported cryo-EM structures of EF-G-bound complexes with a frameshifting-prone mRNA, which suggest that +1 frameshifting can occur before completion of the 30S head swivel[18]. In the current work, domain 4 of EF-G interacts with both the codon and the anticodon in Structures IV and V (Supplementary Fig. 6f–h). They demonstrate that EF-G must remain bound to the ribosome until achieving the latest head-swiveled intermediate Structure V with the most translocated

tRNAs, to support the tRNA-mRNA helix and prevent frameshifting.

Together with recent time-resolved cryo-EM studies of translation initiation, mRNA decoding, termination, and recycling, our work offers a more complete structural visualization of the ribosomal translation cycle. Consistent with biochemical studies, the structural studies revealed that similar inherent and spontaneous ribosomal dynamics (e.g., intersubunit rotation, 30S head swivel) are essential for each step of translation, and that translation factors provide checkpoints that promote accuracy and directionality. These structural dynamics are similar between bacterial systems (*E. coli* and *Th. thermophilus* being the predominant model systems), yeast and mammalian cytosolic and mitochondrial ribosomes, in keeping with the central role of ribosomal RNA and universal conservation of the two-subunit and subunit-domain architecture of the ribosome.

## Methods

**Preparation of EF-G and ribosomal subunits**. The gene encoding the full-length C-terminally His$_6$-tagged *E. coli* EF-G was cloned into a pET24a+ vector (Novagen, kanamycin resistance), and the plasmid was transformed into *E. coli* BLR/DE3 cells. The cells were cultured in Luria-Bertani (LB) medium with 50 μg mL$^{-1}$ kanamycin at 37 °C until the OD$_{600}$ of 0.7–0.8. Expression of EF-G was induced by 1 mM IPTG (Gold Biotechnology Inc., USA), followed by cell growth for 9 h at 16 °C. The cells were harvested, washed and resuspended in buffer A: 50 mM Tris•HCl (pH 7.5), 50 mM NH$_4$Cl, 10 mM MgCl$_2$, 5% glycerol, 10 mM imidazole, 6 mM β-mercaptoethanol (βME) and protease inhibitor (complete Mini, EDTA-free protease inhibitor tablets, Sigma Aldrich, USA). The cells were disrupted with a microfluidizer (Microfluidics, USA), and the soluble fraction was collected by centrifugation using a JA-20 rotor at 39,200 × g for 50 minutes and filtered through a 0.22 μm pore size sterile filter (CELLTREAT Scientific Products, USA).

EF-G was purified in three steps. The purity of the protein after each step was assessed by 12% SDS-PAGE stained with Coomassie Brilliant Blue R 250 (Sigma-Aldrich). First, affinity chromatography with Ni-NTA column (Nickel-nitrilotriacetic acid, 5 ml HisTrap, GE Healthcare) was performed using FPLC (Äkta explorer, GE Healthcare) at 4 °C. The cytoplasmic fraction was loaded onto the column equilibrated with buffer A and washed with the same buffer. EF-G was eluted with a linear gradient of buffer B (buffer A with 0.25 M imidazole). Fractions containing EF-G were pooled and dialyzed against buffer C (50 mM Tris•HCl (pH 7.5), 100 mM KCl, 10 mM MgCl$_2$, 0.5 mM EDTA, 6 mM βME and protease inhibitor). The second purification step involved ion-exchange chromatography using a 20-ml HiPrep FF Q-column (GE Healthcare). The column was equilibrated and washed with buffer C. EF-G sample was loaded in buffer C and eluted with a linear gradient of buffer D (buffer C with 0.7 M KCl). Finally, the protein was dialyzed against 50 mM Tris•HCl (pH 7.5), 100 mM KCl, 10 mM MgCl$_2$, 0.5 mM EDTA, 6 mM βME, and purified using size-exclusion chromatography (Hiload 16/600 Superdex 200 pg column, GE Healthcare). The fractions of the protein were pooled, buffer-exchanged (25 mM Tris•HCl (pH 7.5), 100 mM NH$_4$Cl, 10 mM MgCl$_2$, 0.5 mM EDTA and 6 mM βME, 5% glycerol) and concentrated with an ultrafiltration unit using a 10-kDa cutoff membrane (Millipore). The concentrated protein was flash-frozen in liquid nitrogen and stored at −80 °C.

70S ribosomes were prepared from *E. coli* (MRE600), and stored in the ribosome-storage buffer (20 mM Tris•HCl (pH 7.0), 100 mM NH$_4$Cl, 12.5 mM MgCl$_2$, 0.5 mM EDTA, 6 mM βME) at −80 °C[18]. Ribosomal 30S and 50S subunits were purified using sucrose gradient (10–35%) in a ribosome-dissociation buffer (20 mM Tris•HCl (pH 7.0), 500 mM NH$_4$Cl, 1.5 mM MgCl$_2$, 0.5 mM EDTA, 6 mM βME). The fractions containing 30S and 50S subunits were collected separately, concentrated and stored in the ribosome-storage buffer at −80 °C.

**Preparation of charged tRNAs and mRNA**. Native *E. coli* tRNA$^{fMet}$ was purchased from Chemical Block and was aminoacylated as described (Lancaster and Noller, 2005). Native *E. coli* tRNA$^{Pro}$ (UGG) was over-expressed in *E. coli* from an IPTG-inducible *proM* gene encoded by the pKK223-3 plasmid. Total tRNA was isolated using differential centrifugation and tRNA$^{Pro}$(UGG) was isolated using a complementary biotinylated oligonucleotide attached to streptavidin-sepharose, yielding approximately 40 nmoles tRNA$^{Pro}$(UGG) from 1 liter of culture. tRNA$^{Pro}$ (UGG) (10 μM) was aminoacylated in the charging buffer (50 mM Hepes (pH 7.5), 50 mM KCl, 10 mM MgCl$_2$, 10 mM DTT) in the presence of 40 μM L-proline, 2 μM prolyl-tRNA synthetase, 0.625 mM ATP and 15 μM elongation factor EF-Tu. EF-Tu was purified as described[47]. The mixture was incubated for 10 minutes at 37 °C. To stabilize the charged Pro-tRNA$^{Pro}$ in the form of Pro-tRNA$^{Pro}$•EF-Tu•GTP ternary complex, 0.25 mM GTP was added to the mixture. The mixture was incubated for 3 minutes at 37 °C.

Model mRNA, containing the Shine-Dalgarno sequence and a linker to place the AUG start codon (underlined) in the P site and proline (bolded) in the A site (GGC AAG GAG GUA AAA AUG CCA AGU UCU AAA AAA AAA AAA) was synthesized by IDT.

**Preparation of the 70S translocation complex with EF-G•GTP**. The 70S•mRNA•fMet-tRNA$^{fMet}$•Pro-tRNA$^{Pro}$•EF-G•GTP reactions were prepared as follows. First, a pre-translocation complex with fMet-Pro-tRNA$^{Pro}$ in the A site was assembled. 0.33 μM 30S subunit (all concentrations are specified for the final solution) were pre-activated at 42 °C for 5 minutes in the ribosome-reconstitution buffer (20 mM HEPES (pH 7.5), 120 mM NH$_4$Cl, 20 mM MgCl$_2$, 2 mM spermidine, 0.05 mM spermine, 6 mM βME). 0.33 μM 50S subunit with 1.33 μM mRNA were added to the 30S solution and incubated for 10 minutes at 37 °C. To form the 70S initiation-like complex, 0.33 μM fMet-tRNA$^{fMet}$ was added, and the solution was incubated for 3 minutes at 37 °C. To deliver Pro-tRNA$^{Pro}$ to the A site, the pre-incubated ternary complex (Pro-tRNA$^{Pro}$ at 0.33 μM; EF-Tu at 0.5 μM; GTP at 0.25 mM) was added to the solution and incubated for 10 minutes at 37 °C, as described[18].

Translocation complexes with EF-G were formed by addition of the mixture of ice-cooled 5.3 μM EF-G and 0.66 mM GTP to the ice-cooled pre-translocation complex, on ice. No EF-G and GTP were added to the 0-s time point reaction, which was applied to a grid and blotted as described below. The 10-μL reaction with EF-G was mixed and an aliquot was immediately applied on the grid, blotted and plunged into a cryogen, as described below, resulting in the 25-s time-point sample. The 3600-s sample was obtained by incubation of the pre-translocation complex with EF-G and GTP for 60 minutes on ice followed by grid blotting and plunging.

To form a viomycin-bound pre-translocation complex (Structure III-vio), 0.13 mM viomycin was added to the pre-translocation complex and incubated for 3 minutes at 37 °C. 5.3 μM EF-G and 0.66 mM GTP were added to the solution, incubated for 5 minutes at 37 °C, cooled down to room temperature, applied to a grid and plunged into a cryogen.

**Cryo-EM grid preparation, data collection, and image processing**. QUANTI-FOIL R 2/1 grids with 2-nm carbon layer (Cu 200, Quantifoil Micro Tools) were glow discharged with 25 mA with negative polarity for 60 s in a PELCO easiGlow glow discharge unit. 2.5 μl of each complex was separately applied to the grids. Grids were blotted at blotting force 10 for 4 s at 5 °C, 95% humidity, and plunged into liquid ethane using a Vitrobot MK4 (FEI). Grids were stored in liquid nitrogen.

*25-s dataset* — Data collection and processing of all datasets were performed similarly to those for the 25-s data set (Supplementary Fig. 1 and Table 1), with differences outlined below. Cryo-EM data were collected at the Cryo-EM Center, University of Massachusetts Medical School. From a grid with the 70S•mRNA•fMet-tRNA$^{fMet}$•Pro-tRNA$^{Pro}$ complex that was cryogen-plunged 25 s after mixing with EF-G•GTP, 4,943 movies were collected on a Titan Krios microscope operating at 300 kV (FEI/ThermoFisher) equipped with a K3 Summit camera system (Gatan), with the defocus range of −0.8 to −2.0 μm. Multi-shot multi-hole data acquisition was performed by recording four shots per grid hole from four holes at a time[78], using SerialEM (vs. 3.6)[79] with beam-image shift. Each exposure was acquired with continuous frame streaming at 33 frames per 1.977 s yielding a total dose of 47.58 e$^-$/Å$^2$. The dose rate was 16.54 e$^-$/pixel/s at the camera. The nominal magnification was 105,000 and the calibrated super-resolution pixel size at the specimen level was 0.415 Å. The movies were motion-corrected, and frame averages were calculated using all frames within each movie, after multiplying by the corresponding gain reference in IMOD (vs. 4.9.0)[80]. During motion correction, the movies were binned to the pixel size of 0.83 Å (termed unbinned or 1× binned). cisTEM (vs. 1.0-beta)[81] was used to determine defocus values for each frame average and to pick ribosome particles. 238 movies with large drift, low signal, heavy ice contamination, or very thin ice were excluded from further analysis after inspection of the averages and the power spectra computed by CTFFIND4 within cisTEM. The stack of 475,746 particles and particle parameter files were assembled in cisTEM with the binnings of 1×, 2×, 4×, and 8× and the box size of 448 unbinned pixel[3]. FREALIGNX was used for particle alignment, refinement and final reconstruction steps. FREALIGN v9.11 was used for 3D classification steps[82], as shown in Supplementary Fig. 1. The 8x-binned image stack was initially aligned to a 70S ribosome reference (PDB 5U9F)[83] using 5 cycles of mode 3 alignment (global search), including data in the resolution range of 300–30 Å until the convergence of the average score. Subsequently, the 8× binned stack was aligned against the common reference resulting from the previous step, using mode 1 (refine) in the resolution range 300–18 Å (3 cycles of mode 1). The 2× binned image stack was then aligned against the common reference using mode 1 (refine) in several steps, in which the high-resolution limits gradually increased to 8 Å (5 cycles per each resolution limit). 3D density reconstruction was obtained using 60% particles with the highest scores. Subsequently, the refined particle parameters were used for classification of the 2× binned stack into 16 classes in 100 cycles, using the resolution range of 300–8 Å. This classification revealed 9 high-resolution classes, 4 low-resolution (junk) classes, and 3 classes representing only 50S subunit (Supplementary Fig. 1a). The particles assigned to the high-resolution classes were extracted from the 2× binned stack (with > 50% occupancy and > 0 score) using merge_classes.exe (part of FREALIGN distribution), resulting in the stack of 262,085 particles. Classification of this stack was performed for 100 cycles

using a spherical mask (40 Å radius) focused to cover most of the ribosomal A and P sites. Classification into 16 classes yielded two 70S maps, each of which contained densities for two tRNAs and EF-G. To better resolve the positions of tRNAs and EF-G, these maps were subject to additional classification. To this end, the particles assigned to these two classes were extracted from the 2× binned stack into two sub-stacks (with > 50% occupancy and scores > 0) using merge_classes.exe, resulting in stack-1 and stack-2 with 7173, and 12,327 particles, respectively. Prior to classification, stack-1 was re-refined, as described above, using the high-resolution cutoff of 6 Å. Classification was performed for 100 cycles, using the same A site focused mask. Classification yielded into 2 classes yielded a 3.3 Å class (Structure IV) which contained both tRNAs and EF-G, and a class containing a rotated ribosome with P/E tRNA. An additional classification of stack-1 using a separate masking strategy was performed (Supplementary Fig. 1b). Classification of stack-1 was performed for 100 cycles, using a focused spherical mask with the radius of 35 Å, covering the GTPase domain of EF-G. Classification into 2 classes yielded a 3.3 Å class (Structure IV$_{gtpase}$) which contained both tRNAs and EF-G, and a heterogenous class requiring further classification. Subsequent classification of the second class into 2 classes using a focused mask of 30 Å around the 50S E site produced a 3.9 Å class with two tRNAs and EF-G (Structure V), and a heterogenous class requiring additional classification. A final classification of the heterogenous class for 100 cycles with a 35-Å spherical focused mask around the translocase domain of EF-G produced a 3.8 Å class that contained the 70S ribosome with a swiveled head, two tRNAs, and no EF-G (Structure VI). Stack-2 exhibited heterogeneity at the EF-G binding site and 30S domain conformation, so it was first classified into 2 classes for 100 cycles, using a 3D mask covering the shoulder domain, filtered to 30 Å and down-weighted to 0.1. The class containing EF-G was used to create a subset of 5,379 particles (stack-2a) which was subjected to a 100-cycle classification into 3 classes using the original 40 Å A site mask. This classification produced a 3.8 Å class of 1,657 particles containing a pre-translocation 70S•tRNA•EFG state (Structure III). An additional classification of stack-2a using a separate masking strategy was performed (Supplementary Fig. 1c). Classification of stack-2a was performed for 100 cycles, using a focused spherical mask with the radius of 35 Å, covering the GTPase domain of EF-G. Classification into 3 classes yielded a 3.7 Å class of 1,884 particles containing a pre-translocation 70S•2tRNA•EFG state (Structure III$_{gtpase}$).

*0-s dataset* — Data collection and processing for the 0-s time point pre-translocation 70S•mRNA•fMet-tRNA$^{fMet}$•Pro-tRNA$^{Pro}$ complex were performed as follows. Two data sets containing 1,161 and 4,217 movies were collected with nearly identical parameters on the Titan Krios microscope described above, with the −0.5 to −1.5 μm defocus range. Multi-shot multi-hole data acquisition was performed using SerialEM as described above. Each exposure was acquired with continuous frame streaming at 25 frames per ~1 s yielding a total dose of ~40 e⁻/Å² (Supplementary Table 1). The movies were motion-corrected and frame averages were calculated using all frames in IMOD. The nominal magnification was 105,000 and the calibrated super-resolution pixel size at the specimen level was 0.415 Å. During motion-correction, the movies were binned to the pixel size of 0.83 Å (termed unbinned or 1× binned). The initial alignment, refinement and 3D classification of both stacks of 137,421 (stack-1) and 686,834 (stack-2) particles into 16 classes for 100 cycles was performed, as described for the 25-s dataset above, with the exception of an ab initio model generated from 50% of the particles of stack-1 was used for initial alignment. After excluding the low-resolution (junk) classes and classes representing the 50S subunit, the extracted particles were combined into 2×-binned substack-1 (91,638 particles) and substack-2 (346,334 particles, respectively) and classified into 16 classes using a 40-Å focused spherical mask placed between the A and P sites (as in the 25-s data set) to resolve the pre-translocation classes. Seven classes from substack-1 containing non-rotated pre-translocation ribosomes were combined (37,252 particles) and refined resulting in Structure I. 5 classes of rotated ribosomes from substack-1 and 2 classes of rotated ribosomes from substack-2 were combined into a new stack of 122,977 particles (stack-3). Initial alignment and refinement were repeated on stack-3. 2x-binned stack-3 was classified into 24 classes using the same A site 40-Å focused spherical mask. Singles classes of 7,257 and 6,895 particles containing rotated pre-translocation ribosomes with P/E tRNA and either A/A or A/P* tRNA were separated and refined resulting in Structures II-A and II-B, respectively. 6 classes from stack-3 which contained A-site density were combined into a substack of 22,731 to be further inspected for the anticipated A/P and P/E tRNA state. Several masking strategies could not isolate this state and we suspect it exists in very low abundance due to the unique tRNA dynamics of Pro-tRNA$^{Pro}$.

*3600-s dataset*—2574 movies were collected for the 3600-s 70S•mRNA•fMet-tRNA$^{fMet}$•Pro-tRNA$^{Pro}$•EF-G•GTP complex on a Talos Arctica microscope operating at 200 kV (FEI), equipped with a K3 Summit camera system (Gatan), with the defocus range of −0.5 to −1.5 μm. Multi-shot data collection was performed by recording shots from four holes at a time, using SerialEM, as described above. Each exposure was acquired with continuous frame streaming at 27 frames per 1.13 s, yielding a total dose of 30.4 e⁻/Å². The dose rate was 16.48 e⁻/pixel/s at the camera. The movies were motion-corrected and frame averages were calculated using all frames within each movie after multiplying by the corresponding gain reference in IMOD. The nominal magnification for the dataset was 45,000 and the calibrated super-resolution pixel size at the specimen level was 0.435 Å. During motion correction, the movies were binned to the pixel size of 0.87 Å. The initial alignment, refinement and 3D classification of the original

170,799-particle stack into 16 classes were performed as described for the 25-s dataset. After excluding the low-resolution classes and classes representing the 50S subunit, the extracted particles were combined into a 2x-binned substack (132,070 particles) and classified into 16 classes using a 40-Å focused spherical mask positioned between the A and P sites, as in the 0- and 25-s data sets. Post-translocation states were combined into a single stack of 55,457 particles and refined using a 6 Å resolution cutoff, resulting in the 2.9 Å map (Structure VII).

We had sufficient resolution to identify the P-site codon. Particles were assigned as substrate (70S containing A-site Pro-tRNA$^{Pro}$), EF-G bound intermediate, and product (70S containing P-site Pro-tRNA$^{Pro}$) in our quantification of all data sets.

**Dataset for the viomycin-bound complex**. For the viomycin-bound 70S•mRNA•fMet-tRNA$^{fMet}$•Pro-tRNA$^{Pro}$•EF-G•GTP pre-translocation complex, a dataset of 4,740 movies was collected on a Talos Arctica microscope operating at 200 kV (FEI) equipped with a K3 Summit camera system (Gatan), with the defocus range of −0.5 to −1.5 μm. Multi-shot data collection was performed by recording shots from four holes at a time, using SerialEM, as described above. Each exposure was acquired with continuous frame streaming at 27 frames per 1.618 s yielding a total dose of 30.48 e⁻/Å². The dose rate was 14.30 e⁻/pixel/s at the camera. The nominal magnification was 45,000 and the calibrated super-resolution pixel size at the specimen level was 0.435 Å. The movies were motion-corrected and frame averages were calculated using all frames within each movie after multiplying by the corresponding gain reference in IMOD. During motion correction in IMOD the movies were binned to pixel size 0.87 Å (termed unbinned or 1× binned). cisTEM was used to determine defocus values for each frame average and for particle picking. 114 movies with large drift, low signal, heavy ice contamination, or very thin ice were excluded from further analysis after inspection of the averages and the power spectra computed by CTFFIND4 within cisTEM. The stack of 517,847 particles and the particle parameter files were assembled in cisTEM with the binnings of 1×, 2×, 4×, and 8× and the box size of 448 unbinned pixel. Particle alignment and refinement against the the 8×-binned and 2×-binned stacks were performed as described for the 25-s dataset. The 2× binned stack was classified into 16 classes in 50 cycles, using the resolution range of 300–8 Å. This classification revealed 9 high-resolution classes, 5 low-resolution (junk) classes and 2 classes representing the 50S subunit (Supplementary Fig. 2). Particles assigned to the high-resolution classes were extracted from the 2× binned stack (with > 50% occupancy and scores > 0) using merge_classes.exe (part of the FREALIGN distribution), and merged into a stack containing 322,549 particles. Classification of this stack was performed for 50 cycles using a focused spherical mask with the 30-Å radius, covering most of the A and P sites. Classification into 16 classes yielded 3 high-resolution classes, each of which contained two tRNAs and EF-G. The particles assigned to the 3 high-resolution classes were extracted from the 2× binned stack (with > 50% occupancy and scores > 0) using merge_classes.exe (part of the FREALIGN distribution), and merged into a stack containing 48,345 particles. Classification of this stack was performed for 50 cycles using a focused spherical mask at the A site (30 Å radius, as implemented in FREALIGN). Classification into 3 classes yielded a single high-resolution class, which contained two tRNAs and EF-G. Additional classification of each class into more classes did not yield other unique high-resolution structures with EF-G. For the class of interest (Structure III-vio, 20,167 particles), particles with > 50% occupancy and scores > 0 were extracted from the 2× binned stack. Refinement from the 2× binned stack. Refinement to 6 Å resolution using mode 1 (5 cycles) of the 1× binned stack using 95% of particles with highest scores resulted in a 3.4 Å map (FSC = 0.143). Beamtilt correction to the Nyquist limit (beamtilt(x,y) mrad = −0.0198 −0.0290; particle shift x,y (A) = 0.0849, 0.1189) has further improved map quality, yielding a 3.2 Å map (FSC = 0.143).

**Map filtering and resolution**. Local-resolution filtering was applied to the resulting cryo-EM maps by a previously optimized procedure[36], using blocres and blocfilt from the Bsoft (vs. 1.9.1) package[84], followed by sharpening the blocfiltered maps with bfactor.exe using a constant B-factor of −50 Å² to the average resolution determined by FSC_part. These maps were used for model building and structure refinements. Maps sharpened or softened with different B-factors (from −125 to +50 Å²) were also used to interpret high-resolution details or lower-resolution features. FSC curves were calculated by FREALIGN for even and odd particle half-sets (Supplementary Figs. 1–2).

**Model building and refinement**. Cryo-EM structure of E. coli 70S•fMet-tRNA$^{Met}$•Phe-tRNA$^{Phe}$•EF-Tu•GDPCP Structure III[47], excluding EF-Tu and tRNAs, was used as a starting model for structure refinements. The structures of EF-G were created by homology modeling and map fitting, using ribosome-bound EF-G structures including PDB 4V7D (Brilot et al. 2013, PNAS), PDB 7K51, PDB 4V9H, PDB 4W29[13] and the crystal structure of EF-G-2 (PDB 1WDT) as references. Initial protein and ribosome domain fitting into cryo-EM maps was performed using Chimera[85], followed by manual modeling using Pymol (vs. 1.7.x)[86]. The linkers between the domains and parts of the domains that were not well defined in the cryo-EM maps (e.g. ribosomal proteins or loops of EF-G) were not modeled.

Atomic models were refined against corresponding cryo-EM maps by real-space simulated-annealing refinement using atomic electron scattering factors in RSRef (2000)[87]. Secondary-structure restraints, comprising hydrogen-bonding restraints

for ribosomal proteins and base-pairing restraints for RNA molecules, were employed as described[88]. Refinement parameters, such as the relative weighting of stereochemical restraints and experimental energy term, were optimized to produce stereochemically optimal models that closely agree with the corresponding maps. In the final stage, the structures were refined using phenix.real_space_refine (vs. 1.14_3260)[89], followed by a round of refinement in RSRef applying harmonic restraints to preserve protein backbone geometry while improving RNA geometry, and lastly by atomic B-factor refinement in phenix.real_space_refine. The refined structural models closely agree with the corresponding maps, as indicated by low real-space R-factors of ~0.25 and Correlation Coefficients of ~0.75 (Supplementary Table 1) and visual inspection of the models and maps. The resulting models have excellent stereochemical parameters as indicated by the low MolProbity scores of ~2, low deviations from ideal bond lengths and angles, low number of protein and RNA outliers and other structure-quality statistics (Supplementary Table 1). Structure quality was validated using MolProbity.

Structure superpositions and distance calculations were performed in Pymol. To calculate an angle of the 30S body or head rotation with respect to Structure VII, 23S rRNA (body rotation) or the 16S rRNA excluding the head domain (i.e. residues 2–920 and 1398–1540; head rotation) were aligned with the corresponding rRNA from Structure VII using PyMOL, and the angles between 16S body or head regions were measured in Chimera (vs. 1.13). Figures were prepared in PyMOL, GraphPad Prism 8, and Chimera[85,86]. Buried surface area (contact area) for EF-G domains was calculated in Pymol using the refined EF-G structures (III, IV, and V) with generated hydrogen atoms in the presence and absence of the ribosome components.

**Assessment of GTP and GDP•Pi models fit into cryo-EM maps.** To differentiate between the stages of GTP hydrolysis in the EF-G GTPase center, we quantitatively assessed the local fit of the alternative structural models (pre-hydrolysis GTP or post-hydrolysis GDP•Pi) into density maps for pre-translocation states in this work (III and III-vio) and in the recent study that captured ribosome•EF-G with spectinomycin and reported a structural model with GTP[65]. The cryo-EM map from the latter study has a visually better fit for the γ-phosphate separated from the β-phosphate (Supplementary Figs. 6l–n), and the structural model for GTP contained stereochemical outliers indicating overfitting (e.g. RMS deviation from ideal bond lengths of ~0.03 Å and RMS deviation from ideal bond angles of ~6°; Supplementary Table 3). To this end, the original nucleotide models were used as starting models. In addition, high-resolution reference models of GTP and GDP•Pi from a 1.5 Å crystal structure of aIF2 (PDB 4RD1; termed aIF2 nucleotide) were fit into the maps by aligning the GTPase domain of aIF2 onto EF-G GTPase. B-factors were set to a uniform value (80) for all structural models prior to refinement. The corresponding maps were carved around the modeled nucleotide using phenix.map_box_1.14-3260. The nucleotides were refined using phenix.real_space_refine at an optimal weight resulting in a good fit of the nucleotide/Pi into the map and good stereochemical parameters, as reported in Supplementary Table 3. With all three maps and different starting models, GDP•Pi structural models produced a superior fit over GTP.

**Reporting summary.** Further information on research design is available in the Nature Research Reporting Summary linked to this article.

## Data availability
The data that support this study are available from the corresponding author upon reasonable request. The EM density maps generated in this study have been deposited in the EMDB under accession codes EMD-25420 (Structure I); EMD-25411 (Structure II-A); EMD-25410 (Structure II-B); EMD-25409 (Structure III); EMD-25407 (Structure IV); EMD-25405 (Structure V); EMD-25415 (Structure VI); EMD-25418 (Structure VII); EMD-25421 (Structure III-vio). The atomic coordinates generated in this study have been deposited in the PDB under accession codes 7ST6 (Structure I); 7SSO (Structure II-A); 7SSN (Structure II-B); 7SSL (Structure III); 7SSD (Structure IV); 7SS9 (Structure V); 7SSW (Structure VI); 7ST2 (Structure VII); 7ST7 (Structure III-vio).

Structures from prior studies were used in this work for comparison and are available in the Protein Data Bank: 6WDF, 5U9F, 4V7D, 7K51, 4V9H, 4W29, 1WDT, 4RD1, 5UYM, 4WPO, 7N2V, 4V7B, 3J9Z, 4V5F, 7K52, 2DY1, and 2EFG. Additionally, the corresponding map for PDB 7N2V (EMD-24134) was used in refinement analyses (Methods).

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

## Acknowledgements

We thank Chen Xu and Kangkang Song for grid screening and data collection at the cryo-EM facility at UMass Chan Medical School; Darryl Conte Jr. and members of the Korostelev laboratory for discussions and comments on the manuscript. This study was supported by LL2008 project with the financial support of MEYS CR as part of the ERC CZ program and by Czech Science Foundation, project no. GJ20-16013Y (to G.D.), and by NIH Grants F31 HL152650 (to C.E.C.), R35 GM134931 (to Y.M.H.) and R35 GM127094 (to A.A.K.).

## Author contributions

Conceptualization: G.D., A.A.K. Methodology: C.E.C., G.D., A.B.L., H.G., Y.M.H., A.A.K. Validation: C.E.C., G.D., A.A.K. Investigation: C.E.C., G.D. Resources: Y.M.H., G.D., A.A.K. Writing- Original Draft: C.E.C., A.A.K. Writing- Review and Editing: All; Visualization: C.E.C., A.B.L., G.D., A.A.K. Supervision: A.A.K. Funding Acquisition: C.E.C., Y.M.H., G.D., A.A.K.

## Competing interests

The authors declare no competing interests.
