## [Peer Review File · Nature Communications]

Time-resolved cryo-EM visualizes ribosomal translocation with EF-G and GTPReviewers' Comments:

Reviewer #1:

Remarks to the Author:

The manuscript by Carbone et al. "Time-resolve cryo-EM visualizes ribosomal translocation with EF-G and GTP" reports nine 70S ribosome structures in three distinct states of mRNA-tRNA translocation. The authors used time-resolve cryo-EM to capture the ribosome in the process of tRNA translocation in the presence of EF-G and GTP. The reaction mixture containing ribosomes and tRNAs was prepared on ice without EF-G (0 sec time point) and with EF-G freezing grids at two time points (25 sec and 3600 sec). The nine ribosome reconstructions were classified based on the state of tRNA binding and the presence/absence of bound EF-G. The structures fall into three groups: pre-translocation state, EF-G-bound intermediates, and post-translocation state. The main findings are that nearly rigid EF-G rectifies the intrinsic ribosome movements to promote directional tRNA-mRNA translocation, and hydrolysis of GTP and Pi release promote dissociation of EF-G.

The most interesting structures are those with bound EF-G in intermediate states of translocation. The data show that extended EF-G binds to the rotated pre-translocation ribosome, as observed from numerous crystal and cryo-EM structures obtained with inhibitors or non-hydrolysable GTP analogs. The authors report three structures bound to EF-G, which appear to recapitulate tRNA-mRNA progression through the ribosome. The progression of the reaction can also be deduced from the state of G-nucleotide bound and the density of the G-domain and domain 2. Structure III (first intermediate with EF-G) features the ribosome in a rotated conformation with the head swiveled by about 3.5 degrees. In this structure, EF-G is bound to GDP and the density suggests that Pi was also trapped in the nucleotide binding pocket. The authors performed cryo-EM with a similar complex in the presence of viomycin and observed similar density, and since the resolution was slightly higher (3.2A), the Pi was better resolved.

In the reconstruction of structure III, EF-G is extended and the authors claim that its position (domain 4) is not compatible with that of tRNA in the A/P and A/P* seen structures II-A and II-B. On lines 186-188, the fact that EF-G would clash with the A-site tRNA in structures II-A and II-B, and that EF-G is bound to structure III, does not mean that structure III is "the" substrate for EF-G binding. The authors should re-phrase this, such as "...is the likely substrate for EF-G binding" or "appears to be the preferred substrate...". It is better to exert caution here; the data suggests that structure III is one of the substrates for EF-G binding but may not be the only one. For instance, it was shown by Chen J et al. NSMB 2013 (ref 33 in manuscript) that EF-G samples both the rotated and non-rotated states, binding with higher affinity to the rotated state. To support their findings that EF-G engages the spontaneously rotated ribosome conformation, the authors should refer to Rundlet EJ et al. Nature 2021 (PMID: 34234344). Furthermore, it is unclear how structure III compares to previous EF-G-ribosome complex structures with a non-hydrolysable GTP analog. For instance, in Zhou J et al. Science 2013, the 30S body is rotated by 15-18 degrees and the head swivels by 3-5 degrees, which is very similar to structure III reported here.

In structure IV (second intermediate with EF-G), the ribosome is less rotated but the head swivels to about 17 degrees. In this reconstruction, the A-tRNA has moved by ~20A towards the P site. The authors say that the tRNA binding states closely resemble that seen in a previous structure (Zhou J et al. Science 2014). This needs to be shown in Extended Data Figures.

Structure V (third intermediate with EF-G) reports a highly swiveled 30S head domain (18.1 degrees) and a barely rotated 30S body (1.1 degrees). This state likely represents a late translocation intermediate.

During the transition between the intermediates, the authors see density for the γ -phosphate that is separated from GDP, and attribute it to Pi (structure III). Interestingly, sw1 bridges the 30S body with the GTPase activating center (GAC) on the 50S subunit, suggesting that the rotated 30S subunit stabilizes the active conformation of EF-G. The higher resolution (3.2A) structure with the antibiotic

viomycin appears to suggest that there is indeed density for Pi next to GDP. In structure IV, GDP is bound and the G-domain separates from the sarcin-ricin loop (SRL). Then, in structure V, there is no density attributable to the G-domain and domain 2, indicating they dissociated. Overall, this is a sound and well executed study that reports new aspects and clarifies the mechanism of EF-G-mediated tRNA translocation on the ribosome.

Additional comments:

All along the authors refer to "dipeptidyl-tRNA^{Pro}" but never show whether they see the dipeptide in the EM density.

There have been many ribosome-EF-G structures published in the recent years representing different states of tRNA binding and ribosome rotation/head swiveling. It would be beneficial for the reader to see how the structures reported here compare to the previous ones. This could be done by superimpositions in an Extended Data figure.

Minor:

There are too many references. This is not a review article.

Line 175: "...(measured between codon nucleotide-34..." should be anticodon.

Lines 230-232: not clear, needs re-phrasing.

Line 248, first word in this line is a repeat.

Line 256: ribosomal protein S12... use the universal protein nomenclature, uS12.

Extended Data Fig. 3: panels i, j and k are not clear. It is essentially impossible to discern the bound nucleotide here. Different color should be used for the nucleotide.

Reviewer #2:

Remarks to the Author:

EF-G facilitates the translocation of tRNA-mRNA complex on the ribosome. The mechanism of translocation has been extensively studied for several decades. Numerous structures of translocation intermediates have been obtained from both x-ray crystallography and cryo-EM. Still, key information on the uncoupling of tRNA-mRNA complex from the 30S subunit remains elusive. In this manuscript, Carbone et al. captured structural snapshots of several authentic translocation intermediates of 70S ribosomes with EF-G bound by time-resolved cryo-EM. Together with additional cryo-EM structures of pre-translocation and post-translocation states, these data have depict a continuous 3D visualization of the translocation process. Overall, this manuscript supports a role of EF-G acts as a molecular pawl to guide the directionality of the inherent and spontaneous ribosomal motions (intersubunit rotation and 30S head swiveling) to promote the translocation. The manuscript is well organized, and clearly written. I therefore only have a couple of minor concerns.

1. The claimed 3-Å local resolution of EF-G GTPase center appeared to be overestimated. As shown in Figs 1e, 4a and 4f, the local density appears to be in the range of 3.5-4 Å. The local density of nucleotides in Fig 4a-c seemed to be segmented out using the "Zone or Color zone" tools in Chimera based on the atomic models, and the zigzag borders indicated that some density cannot be well separated from adjacent atoms. Also, ED figure 3 is not informative. In theory, one should display unsharpened map for local resolution display. If a sharpened map is used, the map should be displayed at a higher contour level (to avoid noise densities).

2. The fractions of particles belong to different translocation states in all three reaction time points were listed in Fig 1. However, the details and criteria of particle assignment were absent. The image processing procedures for the datasets of 0s and 3600s, as well as the pre- and post-translocation maps in the 25s dataset should also be described in Methods.

3. Mixed use of figure citation formats (capital vs lower case, Extended Data Fig vs Fig. S). "Fig. 1H

and Fig. S3" in line 97; Table S2 in line 217.

We appreciate the overall positive and constructive comments that helped us improve our manuscript. We address individual comments below.

Reviewer #1 (Remarks to the Author):

Overall, this is a sound and well executed study that reports new aspects and clarifies the mechanism of EF-G-mediated tRNA translocation on the ribosome.

RESPONSE: We thank the reviewer for this assessment.

In the reconstruction of structure III, EF-G is extended and the authors claim that its position (domain 4) is not compatible with that of tRNA in the A/P and A/P* seen structures II-A and II-B. On lines 186-188, the fact that EF-G would clash with the A-site tRNA in structures II-A and II-B, and that EF-G is bound to structure III, does not mean that structure III is “the” substrate for EF-G binding. The authors should re-phrase this, such as “...is the likely substrate for EF-G binding” or “appears to be the preferred substrate...”. It is better to exert caution here; the data suggests that structure III is one of the substrates for EF-G binding but may not be the only one. For instance, it was shown by Chen J et al. NSMB 2013 (ref 33 in manuscript) that EF-G samples both the rotated and non-rotated states, binding with higher affinity to the rotated state.

RESPONSE: We agree and have re-phrased this statement accordingly.

To support their findings that EF-G engages the spontaneously rotated ribosome conformation, the authors should refer to Rundlet EJ et al. Nature 2021 (PMID: 34234344).

RESPONSE: We now refer the manuscript by Rundlet et al. We note, however, that their structure appears to represent a later translocation intermediate than our Structure III. The spectinomycin-stalled structure is similar to one of our previously published 80S*IRES translocation intermediates (reference 43 in manuscript) in a post-DC-unlocking state. We also refer to the recently published work by Fischer and Rodnina groups, which reports an apramycin-bound structure similar to our III-vio.

Furthermore, it is unclear how structure III compares to previous EF-G-ribosome complex structures with a non-hydrolysable GTP analog. For instance, in Zhou J et al. Science 2013, the 30S body is rotated by 15-18 degrees and the head swivels by 3-5 degrees, which is very similar to structure III reported here.

RESPONSE: Ribosome/tRNA conformations in Zhou et al, 2013 are similar to Zhou et al (2014) and to our Structure IV. Zhou et al, 2013 about their structure: “...including large-scale (15° to 18°) rotation of the 30S subunit head and 3° to 5° rotation of the 30S body”

In structure IV (second intermediate with EF-G), the ribosome is less rotated but the head swivels to about 17 degrees. In this reconstruction, the A-tRNA has moved by ~20Å towards the P site. The authors say that the tRNA binding states closely resemble that seen in a previous structure (Zhou J et al. Science 2014). This needs to be shown in Extended Data Figures.

RESPONSE: We now include an additional Extended Data Fig 8, in which we show comparison of Structure IV with Zhou et al, Science 2014.

Additional comments:

All along the authors refer to “dipeptidyl-tRNA^{Pro}” but never show whether they see the dipeptide in the EM density.

RESPONSE: We now show density for the dipeptide in the PTC in Extended Data Fig 4c.

There have been many ribosome-EF-G structures published in the recent years representing different states of tRNA binding and ribosome rotation/head swiveling. It would be beneficial for the reader to see how the structures reported here compare to the previous ones. This could be done by superimpositions in an Extended Data figure.

RESPONSE: In the new Extended Data Fig 8, we now show comparisons of all three EF-G bound structures with those published previously.

Minor:

There are too many references. This is not a review article.

RESPONSE: We reduced the number of references.

Line 175: “...(measured between codon nucleotide-34...” should be anticodon.

RESPONSE: Thank you for catching this error. Fixed.

Lines 230-232: not clear, needs re-phrasing.

RESPONSE: We rephrased this sentence.

Line 248, first word in this line is a repeat.

RESPONSE: Thank you. Fixed.

Line 256: ribosomal protein S12... use the universal protein nomenclature, uS12.

RESPONSE: Fixed.

Extended Data Fig. 3: panels i, j and k are not clear. It is essentially impossible to discern the bound nucleotide here. Different color should be used for the nucleotide.

RESPONSE: We have updated these panels to emphasize the nucleotide model.

Reviewer #2 (Remarks to the Author):

EF-G facilitates the translocation of tRNA-mRNA complex on the ribosome. The mechanism of translocation has been extensively studied for several decades. Numerous structures of translocation intermediates have been obtained from both x-ray crystallography and cryo-EM. Still, key information on the uncoupling of tRNA-mRNA complex from the 30S subunit remains elusive. In this manuscript, Carbone et al. captured structural snapshots of several authentic translocation intermediates of 70S ribosomes with EF-G bound by time-resolved cryo-EM. Together with additional cryo-EM structures of pre-translocation and post-translocation states, these data have depict a continuous 3D visualization of the translocation process. Overall, this manuscript supports a role of EF-G acts as a molecular pawl to guide the directionality of the inherent and spontaneous ribosomal motions (intersubunit rotation and 30S head swiveling) to promote the translocation. The manuscript is well organized, and clearly written. I therefore only have a couple of minor concerns.

RESPONSE: Thank you.

1. The claimed 3-Å local resolution of EF-G GTPase center appeared to be overestimated. As shown in Figs 1e, 4a and 4f, the local density appears to be in the range of 3.5-4 Å. The local density of nucleotides in Fig 4a-c seemed to be segmented out using the “Zone or Color zone” tools in Chimera based on the atomic models, and the zigzag borders indicated that some density cannot be well separated from adjacent atoms.

RESPONSE: Our local resolution was estimated using the standard Bsoft package broadly used to derive local resolution estimates in cryo-EM maps. Since this point was not clearly articulated in the initial manuscript, we now specify how the resolution was assessed. To be on a conservative side, we now specify the local density of 3.5 Å (rather than 3 Å), as indicated in Extended Fig. 3.

Also, ED figure 3 is not informative. In theory, one should display unsharpened map for local resolution display. If a sharpened map is used, the map should be displayed at a higher contour level (to avoid noise densities).

RESPONSE: We now clarify the B-factor sharpening parameters in the figure legend.

2. The fractions of particles belong to different translocation states in all three reaction time points were listed in Fig 1. However, the details and criteria of particle assignment were absent. The image processing procedures for the datasets of 0s and 3600s, as well as the pre- and post-translocation maps in the 25s dataset should also be described in Methods.

RESPONSE: We have clarified the Methods sections that describe particle number assessments and processing procedures for all three datasets.

3. Mixed use of figure citation formats (capital vs lower case, Extended Data Fig vs Fig. S). “Fig. 1H and Fig. S3” in line 97; Table S2 in line 217.

RESPONSE: Thank you for catching the typos. Fixed.

Reviewers' Comments:

Reviewer #1:

Remarks to the Author:

The authors have addressed this reviewer's concerns.

Minor points:

Line 193, protein L11. This protein is universal, thus "uL11" should be used.

Figure 2, panel f, the gray tRNA (P/E) is not labeled.

Line 570, the bold nucleotides in the mRNA used are not defined.

In Extended Data Fig. 5, it is difficult to discern the different shades of green for A-site tRNA.

Reviewer #2:

Remarks to the Author:

Most of the concerns have been well addressed or clarified. This reviewer has no further comments.

REVIEWERS' COMMENTS

Reviewer #1 (Remarks to the Author):

The authors have addressed this reviewer's concerns.

Minor points:

Line 193, protein L11. This protein is universal, thus "uL11" should be used.

Thank you for catching this. We have corrected this.

Figure 2, panel f, the gray tRNA (P/E) is not labeled.

Addressed.

Line 570, the bold nucleotides in the mRNA used are not defined.

Addressed.

In Extended Data Fig. 5, it is difficult to discern the different shades of green for A-site tRNA.

We updated Supplementary Fig. 5c-d to have more diverse colors so that tRNAs of similar conformation can be seen clearly.

Reviewer #2 (Remarks to the Author):

Most of the concerns have been well addressed or clarified. This reviewer has no further comments.

Thank you for your thoughtful reviews which helped improve our manuscript.